Journal of Machine Learning Research MICCAI COMPAYL (2025) 1-10      Submitted 6/25; Published 10/25

# KidneyGrader: Fine-Grained Tubulitis Scoring Using Weakly Supervised Transformers

**Abrar Rashid**                                                     ARASHID.LDN@GMAIL.COM
*Department of Computing, Imperial College London, UK*

**Vishal Jain**                                                      V.JAIN24@IMPERIAL.AC.UK
*Department of Computing, Imperial College London, UK*

**Sarah Cechnicka**                                     SARAH.CECHNICKA18@IMPERIAL.AC.UK
*Department of Computing, Imperial College London, UK*

**Aamir Chaudry**                                         AAMIR.CHAUDRY20@IMPERIAL.AC.UK
*Department of Computing, Imperial College London, UK*

**Candice Roufosse**                                         CANDICE.ROUFOSSE@NHS.NET
*Centre for Inflammatory Disease, Imperial College London, UK*

**Bernhard Kainz**                                               BKAINZ@IMPERIAL.AC.UK
*Department of Computing, Imperial College London, UK*
*Friedrich–Alexander University Erlangen–Nürnberg, DE*

**Editor:** Linda Studer and Francesco Ciompi

## Abstract

Accurate tubulitis scoring is essential for managing kidney transplant rejection, yet manual assessment is subjective and suffers from severe inter-rater variability ($\kappa_w$=0.17), leading to inconsistent treatment decisions. While recent works have attempted binary tubulitis detection, fine-grained scoring (T0-T3) required for clinical decision-making remains unaddressed. We present the first automated approach for granular tubulitis scoring using only slide-level supervision. Our approach aggregates spatially correlated features from tubule-centric image patches using a transformer-based attention pooling mechanism. To ensure diagnostic focus, patches are pre-filtered using a segmentation model trained to detect renal tubules, restricting the input space to regions most relevant for scoring. Evaluated on 93 routine PAS-stained slides (75 for training/validation, 18 held-out test), our method achieves a weighted kappa of $\kappa_w = 0.75$ (4.4× improvement over expert agreement), 83.3% within-one-grade accuracy, and strong correlation with expert scores ($r = 0.81$). Top-attended regions demonstrate clinical plausibility, showing progressively greater inflammatory burden and tissue damage features with increasing T-scores. Our work demonstrates that weakly supervised learning can transform subjective pathology assessments into reliable, interpretable predictions, offering a practical path towards standardising transplant rejection diagnosis. The code is available on github.

**Keywords:**   Banff Tubulitis Scoring, Inter-Rater Reliability, Reproducibility, Digital Pathology, Deep Learning

# 1 Introduction

Kidney transplant rejection affects 10-15% of recipients within the first year NHS Blood and Transplant (2025) and is diagnosed through Banff lesion scores Roufosse et al. (2018), with tubulitis (T-score) being the primary indicator of acute T-cell mediated rejection Tsamandas et al. (1997). The Banff T-score captures the severity of inflammation within renal tubules, defined as the number of mononuclear leukocytes infiltrating across the tubular basement membrane in the most inflamed tubule. It is graded on a 4-point scale: T0 (none), T1 (1–4), T2 (5–10), and T3 (>10). Such Tubulitis scoring directly determines treatment decisions: T0-T1 cases receive conservative management while T≥2 cases require variants of aggressive immunosuppressive therapy to prevent graft loss. Current manual scoring by expert pathologists achieves extremely poor inter-rater reliability ($\kappa_w$=0.17) Furness and Taub (2001), *i.e.*, pathologists agree barely better than chance when grading the same biopsy. This variability leads to inconsistent treatment decisions, unreliable clinical trial endpoints, and potentially inappropriate therapy, either under-treating rejection or over-immunosuppressing stable patients. This analysis is typically performed on Periodic Acid-Schiff (PAS)-stained sections, a histological stain that highlights carbohydrates such as glycogen, glycoproteins, and basement membranes by producing magenta colouration where these structures are present Aterman and Norkin (1963). In kidney biopsies, PAS staining sharply delineates tubular basement membranes and epithelial structures, enabling pathologists to accurately identify inflammatory cells crossing tubule boundaries.

Tubulitis scoring requires identifying the single most inflamed tubule among hundreds in a gigapixel whole slide image (WSI) ($\sim 64000 \times 64000$ pixels), where inflammatory cells must be counted within tubule boundaries. Prior automation attempts have been limited to binary classification (tubulitis present/absent) Cooper et al., missing the granular T0-T3 distinctions, which are decisive for treatment stratification. Modular pipelines that sequentially segment tubules, detect cells, and compute scores Hermsen et al. (2022) are computationally expensive ($>$ 1 hours per slide), making clinical deployment impractical.

**Related Works.** Previous works have explored kidney allograft histopathology automation, from rejection classification Kers et al. (2022); Ye et al. (2024) to quantitative tissue analysis Hermsen et al. (2022). While these achieve impressive performance for their respective tasks, they do not address the clinical need for granular T-scoring Roufosse et al. (2018). Hermsen et al. (2022) developed a modular pipeline computing tissue metrics that correlate with pathologist scores (Spearman $\rho$=0.838) but stopped short of predicting actual Banff grades. Most recently, Cooper et al. achieved AUC 0.831 for binary tubulitis classification (T0/T1 vs. T2/T3), but this coarse grouping loses the critical distinction between T1 and T2 that determines treatment escalation. To the best of our knowledge, no prior work has attempted fine-grained T0-T3 scoring that preserves clinical decision boundaries while providing explainable predictions.

**Contributions.** We present KidneyGrader, a weakly supervised approach that provides fine-grained T-scores. By incorporating domain knowledge into the learning process, our approach transforms subjective manual scoring into objective, reproducible predictions while maintaining clinical explainability. Our key contributions are:

1. We provide the first automated approach for granular Banff tubulitis scoring (T0-T3) preserving clinical treatment thresholds.

2. By using a novel tubule-guided attention mechanism we achieve 4.4× inter-rater reliability improvement over expert pathologists.

3. Our pilot evaluation on 93 PAS-stained slides (75 train/validation, 18 held-out test) demonstrates $\kappa_w$=0.75, 83.3% within-one-grade accuracy, and strong correlation (r=0.81) with expert labels.

4. We attempt clinical validation through explainable attention visualisations showing appropriate focus on relevant tubuli.

## 2 Method

We formulate tubulitis scoring as a weakly supervised regression task, given that our dataset contains slide-level labels for tubulitis score. Given a gigapixel slide $\mathbf{x} \in \mathcal{X}$ and its slide-level Banff tubulitis grade $y \in \mathcal{Y} = \{0, 1, 2, 3\}$, the goal is to learn a mapping $f : \mathcal{X} \to \mathcal{Y}$ that (i) respects the ordinal structure of $\mathcal{Y}$, (ii) is explainable by highlighting inflammatory tubules, and (iii) requires no patch-level supervision. Each WSI $\mathbf{x}_i$ is decomposed into a bag of patches $\mathcal{B}_i = \{\mathbf{p}_{i,j}\}_{j=1}^{n_i}$ where $\mathbf{p}_{i,j} \in \mathbb{R}^{512 \times 512 \times 3}$ represents the $j$-th patch and $n_i$ is the number of patches in slide $i$. The challenge lies in learning from weak supervision: while we observe slide-level labels $y_i$, the model must identify which patches contain the diagnostically relevant tubular inflammation without explicit patch-level annotations.

**Preprocessing:** We identify tissue regions in each WSI using HSV colour space filtering where pixels satisfy $S(\mathbf{p}) > 0 \wedge V(\mathbf{p}) < 245$, where S and V represent the saturation and value channels respectively. From tissue regions, we extract $512 \times 512$ patches using contiguous grid sampling with 25% overlap. Each patch must contain $\geq$15% tissue content. We apply tubule-content filtering using a 5-class pre-trained attention-gated U-Net with EfficientNet-B0 Cechnicka et al. (2023) Classes represent background (0), tubules (1), glomeruli (2), vessels (3), and interstitium (4). We retain only patches with $\geq$30% tubule content:

$$\mathcal{B}_i^{\text{tubule}} = \{\mathbf{p} \in \mathcal{B}_i \ : \ \text{TubuleRatio}(Mask_{seg}(\mathbf{p})) \geq 0.3\}$$

Then, each patch $\mathbf{p}_{i,j}$ is encoded using the UNI foundation model Chen et al. (2024), $\phi : \mathbb{R}^{512 \times 512 \times 3} \to \mathbb{R}^{1024}$, to produce feature embeddings $\mathbf{h}_{i,j} = \phi(\mathbf{p}_{i,j})$. UNI is a vision transformer pretrained on over 100,000 histopathology images.

**Scaffold:** We base on the TransMIL architecture Shao et al. (2021) with extensions for tubulitis scoring. TransMIL was designed for multi-class classification tasks, but we reformulate it as a regression model, based on (i) better correlation with ground truth compared to classification from empirical observation, and (ii) expert preference for continuous scores that better reflect ambiguous or borderline cases. We process tubule-filtered patches through: (1) feature projection from 1024 to 256 dimensions, (2) 2D positional encoding incorporating spatial coordinates, (3) bidirectional transformer blocks with learnable gating, and (4) class token aggregation for bag-level prediction.

**Bidirectional processing**: We extend standard transformer attention with bidirectional sequence processing. For input features $\mathbf{H} = [\mathbf{h}_1, \ldots, \mathbf{h}_n]$ and coordinates $\mathbf{C} = [\mathbf{c}_1, \ldots, \mathbf{c}_n]$ for n patches, we compute forward and backward attention independently as done in Shao et al. (2021), then combine using learnable gating parameter $\gamma$. This captures both local and distant spatial dependencies, which supports identifying clustered inflammatory patterns.

**T-score regression**: The classification head in Shao et al. (2021) is replaced with a regression module: $\hat{y} = \text{clamp}(\mathbf{w}^T \text{GELU}(\mathbf{W}\mathbf{c}^{(L)} + \mathbf{b}) + b,\ 0,\ 3)$, where $\mathbf{c}^{(L)}$ is the final class token representation, $\mathbf{W}$ and $\mathbf{b}$ are the weights and bias of the projection layer, and $\mathbf{w}$ and $b$ are the weights and bias of the output layer. For instance-level learning, we develop pseudo-labeling where high tubulitis scores ($\geq 2$) generate positive labels for the top-$k$ most attended patches, enabling weak supervision at the patch level.

We also evaluate a regression-adapted variant of the CLAM framework Lu et al. (2020), which uses a gated attention mechanism to aggregate patch features into a slide-level representation $\mathbf{z}$. We replace the original classification head with a regression layer trained using mean squared error loss: $\hat{y} = \mathbf{w}^\top \mathbf{z} + b$.

**Training Objectives**: Both models use a weighted loss: $\mathcal{L} = \alpha\,\mathcal{L}_{\text{bag}} + \beta\,\mathcal{L}_{\text{inst}} + \gamma\,\mathcal{L}_{\text{attn}}$. *Bag-level loss:* CLAM uses mean squared error, while TransMIL uses the Huber loss $H_\delta(\hat{y}, y)$, where $\hat{y}$ is the predicted slide score, $y$ the ground truth, and $\delta$ the transition point between quadratic and linear regimes. *Instance loss:* Both assign binary pseudo-labels $\tilde{y}_j$ to selected patches $j \in \mathcal{S}$ (top-$k$ for CLAM, top-attended for TransMIL), and apply cross-entropy:

$$\mathcal{L}_{\text{inst}} = \sum_{j \in \mathcal{S}} \text{CE}(s_j, \tilde{y}_j),$$

where $s_j$ is the patch-level score. *Attention regularisation:* Used only in TransMIL to promote sparsity:

$$\mathcal{L}_{\text{attn}} = \sum_i \left[ \sum_j a_{i,j} \log(a_{i,j} + \epsilon) + \|\mathbf{a}_i\|_1 \right],$$

where $a_{i,j}$ is the attention weight for patch $j$ in slide $i$, and $\mathbf{a}_i = [a_{i,1}, \ldots, a_{i,n_i}]$.

**Modular baseline:** Since no existing method directly supports fine-grained T-scoring, we automate a manual pathologist workflow step-by-step and construct a modular pipeline: (1) inflammation localisation, (2) tubule instance segmentation, and (3) leukocyte detection with heuristic T-score assignment.

*Stage 1a:* The U-Net mentioned in the preprocessing section segments input patches using a combined Dice and cross-entropy loss. *Stage 1b:* Tubules are isolated via two-pass watershed on the Euclidean distance transform of the tubule mask $\mathbf{M}_{\text{tubule}} = \mathbb{I}[\mathbf{S} = 1]$, where $\mathbf{S}$ is the predicted class mask ($1 = $ tubule), with h-maxima filtering. *Stage 2:* Leukocytes are detected via InstanSeg Goldsborough et al. (2024) at $0.5\,\mu\text{m/pixel}$, followed by ensemble classification using three EfficientNet-B0 models. *Stage 3:* The final score is determined by the maximum inflammatory cell count $c_t$ in any tubule $t$, following Banff grading rules:

$$\hat{y} = \begin{cases} 0 & |\mathcal{T}_{\text{inflamed}}| < 2 \\ 1 & \max_t c_t \leq 4 \\ 2 & 5 \leq \max_t c_t \leq 10 \\ 3 & \max_t c_t > 10, \end{cases}$$

with $\mathcal{T}_{\text{inflamed}} = \{t \in \mathcal{T} \mid c_t > 0\}$ denoting the set of tubules containing inflammatory cells.

## 3 Evaluation & Results

**Dataset**: We evaluate on 93 PAS-stained whole slide images from renal transplant biopsies at Charing Cross Hospital, each annotated with Banff T-scores (T0–T3) by an expert renal pathologist. The dataset contains 20 T0 (21.5%), 24 T1 (25.8%), 23 T2 (24.7%), and 26 T3 (28.0%) cases, providing balance across severity levels. Images were scanned at $40\times$ magnification using a Leica Aperio scanner at 0.263 microns per pixel resolution.

**Training and Evaluation Protocol**: We employ a held-out evaluation protocol with 93 total slides: 18 test slides stratified by class distribution (5 T0, 6 T1, 4 T2, 3 T3) and 75 development slides. For model development, we use 5-fold cross-validation to create an ensemble, with each fold maintaining 60 training and 15 validation slides per stratified split. All five models are evaluated independently on the same 18 held-out test slides, with final prediction for a given method obtained by averaging over the ensemble.

**Data augmentation** is identical for all models: random 90° rotations, horizontal/vertical flips, colour jitter (brightness=0.2, contrast=0.2, saturation=0.2, hue=0.1), and Gaussian noise ($\sigma = 0.1$) with probability 0.5 each.

**TransMIL**: We use our model with two transformer layers, 8 attention heads, embedding dimension 1024, and hidden dimension 256. Dropout is set to 0.3. Training uses the AdamW optimiser (lr=0.001, weight decay=0.01) with cosine annealing scheduler, warmup for 10 epochs, and early stopping (patience=50). Models train for 500 epochs with gradient clipping (value=1.0). Data is processed in MIL-style batches Shao et al. (2021) (batch size=1). The loss function (see Sec. 2) uses $\alpha = 0.8$, $\beta = 0.15$ and $\gamma=0.05$.

**CLAM**: The gated attention network has hidden dimension 128, dropout 0.5, and top-$k$ sampling ($k=32$). Training uses the Adam optimiser (lr=$5\times10^{-5}$, weight decay=0.01) with cosine scheduler, 300 epochs, early stopping (patience = 30), and gradient clipping (value=1.0). The loss (see Sec. 2) uses $\alpha=0.8$, $\beta=0.2$, $\gamma=0$. Label smoothing of 0.1 is applied during training.

**Modular baseline**: (i) The attention-gated U-Net is trained for 60 epochs with AdamW (lr=$3 \times 10^{-4}$); (ii) instance labelling uses a two-pass watershed algorithm with h-maxima transform (h=20); (iii) cell detection is done via the frozen InstanSeg model monkey at $0.5\mu$m/pixel with a classifier ensemble; and (iv) Banff to regularise the model for our small dataset rule-based grading with confidence threshold $p = 0.7$ for inflammatory cells. All methods use PyTorch v2.5 on a single Nvidia RTX A6000 GPU with mixed precision enabled.

**Evaluation Metrics**: Quadratic weighted kappa ($\kappa_w$) enables direct comparison with pathologist inter-rater reliability. Within-1-grade accuracy captures predictions within $\pm1$ grade of expert labels, reflecting acceptable clinical variance. We additionally report Mean Absolute Error (MAE) and Pearson correlation. For explainability assessment, we show clinical plausibility of highly-attended patches across different T-scores.

**Ablation**: We compare two different regression endpoints, adapted TransMIL (ours) vs CLAM to the modular step-by-step baseline in Table 1.

**Results:** We quantify how accurately KidneyGrader predicts Banff T-scores and illustrate *where* the model looks when it makes its prediction. Table 1 benchmarks our full TransMIL variant backbone against the CLAM ablation, the automated modular pipeline, and published pathologist agreement. Across all metrics, with the exception of within-one-grade accuracy, KidneyGrader outperforms the baselines while reducing the inference time from

Table 1: Comparison of tubulitis scoring methods. Best results in **bold**. $\kappa_w$: quadratic–weighted kappa - measures model agreement with expert labels. **Within-1**: percentage of slides whose predicted grade is within $\pm 1$ of the expert label. **MAE**: mean absolute error in Banff grade (lower = better). **Pearson**: linear correlation ($r$) with expert scores. **Exact**: percentage of slides scored identically to the expert. **Runtime**: end-to-end inference time per whole-slide image on a single GPU in minutes. Standard deviation as subscript.

| Method | $\kappa_w \uparrow$ | Within-1 $\uparrow$ | MAE $\downarrow$ | Pearson $\uparrow$ | Exact % $\uparrow$ | Runtime $\downarrow$ |
|---|---|---|---|---|---|---|
| KidneyGrader (Ours) | **0.75** | $83.3_{\pm 6.1}$ | $\mathbf{0.55}_{\pm 0.20}$ | $\mathbf{0.81}_{\pm 0.15}$ | $\mathbf{55.6}_{\pm 9.1}$ | $\mathbf{<5}_{\pm 1}$ |
| KidneyGrader (CLAM) | 0.66 | **100** | $0.66_{\pm 0.13}$ | $0.75_{\pm 0.13}$ | $38.9\%_{\pm 16.0}$ | $<5_{\pm 1}$ |
| Modular Baseline | 0.29 | 82.1 | 0.92 | 0.31 | 28.6 | $\sim 120_{\pm 60}$ |
| Pathologists Furness and Taub (2001) | 0.17 | - | - | - | - | - |

hours to minutes. To enable comparison with existing work and assess performance on the clinically critical treatment threshold, we evaluated binary classification for T≥2 vs. T<2. Table 2 shows that our model achieves an AUC of 0.95, surpassing Cooper et al. despite using a dataset ten times smaller. Table 3 breaks the error down by true class: mis-estimation is largest for subtle T0 cases (MAE 0.70) and smallest for severe T3 (MAE 0.27), reflecting the intuitive difficulty of detecting low-grade inflammation. Figure 1 visualises the most-attended patches for representative slides at each ground-truth grade. For T0–T1 slides the model focuses on intact tubules with little or no inflammatory infiltrate, whereas for T2–T3 slides attention concentrates on tubules densely packed with lymphocytes (highlighted in red). Alignment between saliency and pathology criteria provides visual evidence that the network has learned meaningful morphological cues rather than spurious correlates.

**Discussion:** Our results demonstrate that our method achieves superior tubulitis scoring performance with $\kappa_w = 0.75$, exceeding pathologist inter-rater reliability by 4.7×. This improvement is clinically relevant as quadratic weighted kappa accounts for the ordinal nature of Banff scores and penalises larger disagreements more heavily than adjacent-grade errors. While CLAM achieves perfect within-1-grade accuracy, its lower $\kappa_w$ (0.66) indicates more severe misclassifications when errors occur. TransMIL's balanced performance across both metrics makes it more suitable given the clinical context.

The modular pipeline, despite achieving expert-level reliability ($\kappa_w = 0.29$), is limited by cascading errors from its multi-stage design. The inflammatory cell detector, trained on

Table 2: Binary classification performance (T≥2 vs. T<2). Cooper et al. was trained on a dataset 10× larger than ours.

| Method | AUC↑ | Sens.↑ | Spec.↑ |
|---|---|---|---|
| KidneyGrader | $\mathbf{0.95}_{\pm 0.07}$ | $\mathbf{0.70}_{\pm 0.15}$ | $\mathbf{1.00}_{\pm 0.19}$ |
| Cooper *et al.* | 0.83 | 0.51 | 0.84 |

Table 3: Class-wise mean absolute error (MAE) for the best KidneyGrader fold.

| Metric | T0 | T1 | T2 | T3 |
|---|---|---|---|---|
| MAE↓ | 0.696 | 0.387 | 0.457 | 0.266 |

| **T0** | **T1** | **T2** | **T3** |
|--------|--------|--------|--------|

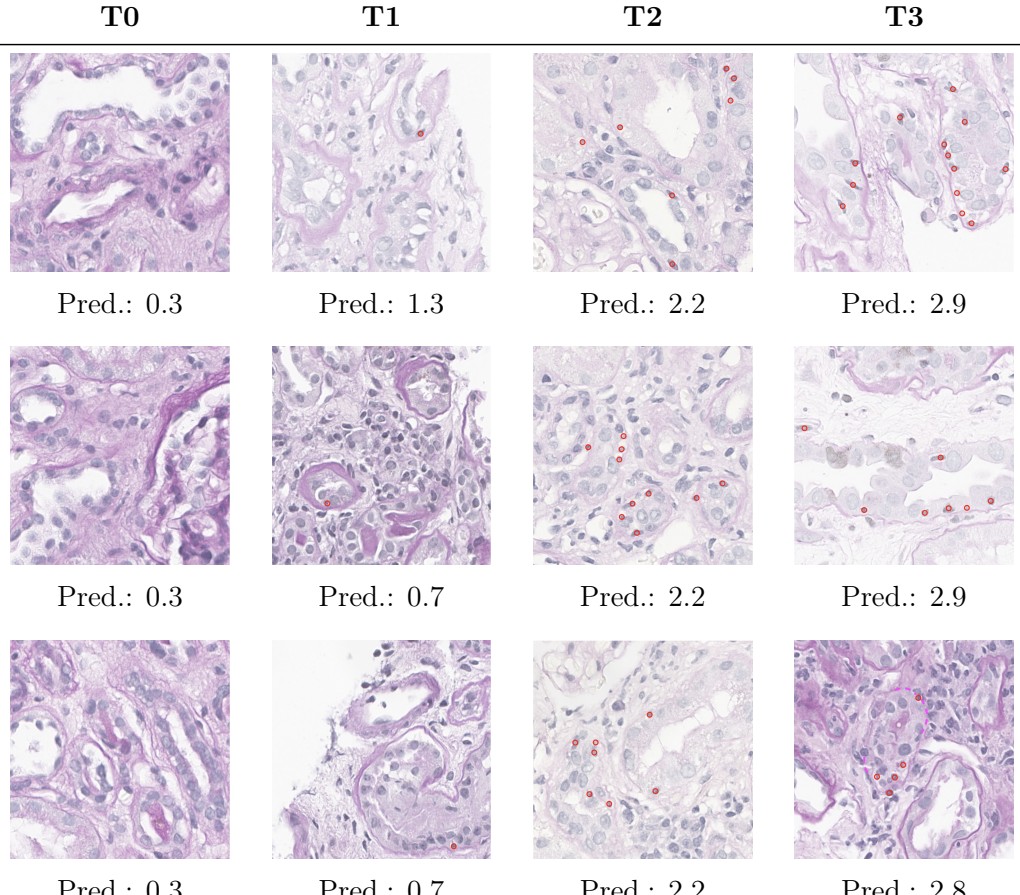

| Pred.: 0.3 | Pred.: 1.3 | Pred.: 2.2 | Pred.: 2.9 |
| Pred.: 0.3 | Pred.: 0.7 | Pred.: 2.2 | Pred.: 2.9 |
| Pred.: 0.3 | Pred.: 0.7 | Pred.: 2.2 | Pred.: 2.8 |

Figure 1: Examples of top attended `KidneyGrader` patches in our test data. Three example patches shown per score category T0-T3, selected from one or more cases. Top-attended patches demonstrate increasing inflammatory burden correlating with T-score: minimal to no inflammation in T0-T1, moderate infiltration in T2, and dense infiltration in T3. One T3 case additionally shows basement membrane dissolution (purple dashed line). Inflammatory cells shown with red markers.

the external MONKEY dataset monkey, suffers from domain shift when applied to inhouse renal biopsies of the same PAS staining, leading to inconsistent cell detection. Additionally, classical instance labelling algorithms struggle with densely packed tubules, creating noisy instance masks that compound downstream errors. Nevertheless, achieving reliability comparable to pathologists demonstrates the viability of two stage method.

Renal tubules frequently span multiple adjacent patches and tubulitis exhibits spatial clustering, requiring contextual information from neighbouring regions. The TransMIL backbone provides better performance than CLAM, likely due to its transformer self-attention enabling explicit patch-to-patch relationships through query-key interactions, whereas CLAM processes patches independently through linear layers. This allows TransMIL to better

capture spatially distributed inflammatory patterns analogous to pathologist workflow of examining clustered inflamed tubules.

Our attention visualisations reveal that the model attends to regions containing features consistent with standard Banff criteria, with high attention on areas showing inflammatory cells and basement membrane dissolution. Notably, detecting basement membrane dissolution in our modular approach would have required additional labelled data, yet here it was implicitly learnt as part of the end-to-end training - demonstrating the advantage of weakly supervised learning. Beyond these Banff-aligned features, we observed the model attending to regions that deviate from standard criteria, including atrophic tubules and integrating inflammatory burden across multiple tubules. Since our model learnt from one pathologist's annotations, these deviations may reflect documented inter-pathologist variability in Banff interpretation Loupy et al. (2022); Mengel et al. (2007).

## 4 Conclusion

We presented the first automated approach for granular Banff tubulitis scoring (T0-T3), addressing a critical need for reproducible assessment in kidney transplant pathology. Our TransMIL-based method with tubule-guided attention achieved $\kappa_w$=0.75 on held-out test data, substantially exceeding reported inter-rater reliability among pathologists ($\kappa_w$=0.17). The approach demonstrated 83.3 % within-one-grade accuracy and strong correlation with expert annotations (r = 0.81), while maintaining clinical efficiency with <5 minute processing time per slide. Our attention analysis demonstrated increasing inflammatory burden correlating with T-score, with the model attending to both Banff-consistent features and regions that deviate from strict criteria. This may reflect the known gap between standardised criteria and clinical interpretation. However, these findings remain preliminary given our limited single-institution, single-pathologist dataset of 93 slides. Future work should validate these results on larger multi-institutional cohorts with consensus annotations from multiple pathologists. Despite current limitations, this work demonstrates the potential for weakly supervised learning to transform subjective pathology assessments into more objective predictions, offering a practical step towards standardising transplant rejection diagnosis.

**Acknowledgments and Disclosure of Funding:** S.C. and V.J. are supported by the UKRI Centre for Doctoral Training AI4Health (EP / S023283/ 1) and (EP/ Y030974 1). Support was also received from the ERC, MIA-NORMAL 101083647, the State of Bavaria (HTA) and DFG 512819079. HPC resources were provided by NHR@FAU of FAU Erlangen-Nürnberg under the NHR project b180dc. NHR@FAU hardware is partially funded by the DFG, 440719683. We thank Imperial's Department of Computing Corporate Partnership Programme for providing travel support. V.J. and C.R. is supported by the National Institute for Health Research (NIHR) Biomedical Research Centre based at Imperial College Healthcare NHS Trust and Imperial College London. The views expressed are those of the authors and not necessarily those of the NHS, the NIHR or the Department of Health. C.R.'s research activity is made possible with generous support from Sidharth and Indira Burman. Human samples used in this research project were obtained from the Imperial College Healthcare Tissue & Biobank (ICHTB). ICHTB is supported by NIHR Biomedical Research Centre based at Imperial College Healthcare NHS Trust and ICL. ICHTB is approved by Wales REC3 to release human material for research (22/WA/2836).

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
