# OpenReview forum: "KidneyGrader: Fine-Grained Tubulitis Scoring Using Weakly Supervised Transformers"
_MICCAI.org/2025/Workshop/COMPAYL — COMPAYL 2025_

### Official Review · Reviewer_413u · 2025-07-11
**Preliminary work on Banff grading with promising clinical implications**

**Rating:** 5
**Confidence:** 3

**Review:**

This paper introduces KidneyGrader, a weakly supervised transformer-based method for fine-grained Banff tubulitis scoring (T0–T3) in kidney transplant biopsies. Although I do not have the technical background to elaborate on the extensive technical details, I found this paper very clinically relevant. It demonstrates substantial improvement in reproducibility compared to expert pathologists, while maintaining fast inference times and offering interpretable outputs through attention maps. Importantly, the authors are transparent about the limitations of their work, clearly positioning it as a first step toward clinical implementation.

Strenghts:
* clinically relevant task
* transparent and self-reflective in discussing limitations of the study
* figure 1 with attention maps for clinicians

Weaknesses:
* as mentioned by the authors: single-institution, single-pathologist, small dataset (n=93)

---

### Official Review · Reviewer_suZY · 2025-07-12
**TransMIL to predict kidney transplant rejection**

**Rating:** 5
**Confidence:** 4

**Review:**

### Summary

The paper tackles the issue of kidney transplant rejection, which affects 15% of recipients within the first year.
The authors use TransMIL to regress the Tubulitis scoring.

### Strenght

- The problem is unique and interesting
- The model is described in detail
- Results are better than baseline

### Weakness

- Models used (TransMIL and CLAM) are old and well known to underperform compared to ABMIL and more recent work.
- I would have appreciated seeing more visual inspections and qualitative analysis to understand the data and the problem better, since it's not a well-known task.

---

### Official Review · Reviewer_Y8E5 · 2025-07-14
**KidneyGrader: Fine-Grained Tubulitis Scoring Using Weakly Supervised Transformers.**

**Rating:** 4
**Confidence:** 5

**Review:**

In this paper, the authors propose KidneyGrader: Fine-Grained Tubulitis Scoring Using Weakly Supervised Transformers. The proposed approach performs tubulitis scoring using weakly supervised slide-level supervision. The authors provide a comprehensive explanation of the clinical background and effectively link it to different treatments that depend on various T-scores (T0 - T1 vs T ≥ 2). Their methodology builds upon TransMIL, a transformer-based model developed for classification. It starts by filtering tubulitis regions using a segmentation model and then feeding the filtered patches to the TransMIL network. The authors frame this as a regression task and replace classification head with a regression head. The model was trained and evaluated on an in-house dataset of 93 slides, with 18 slides held out for testing and the remaining slides used for training (five-fold cross validation). Based on the quantitative reported numbers, their model shows higher kappa agreement in comparison with the baseline. Their attention map analysis also reveals that high-attention regions correspond to areas with increasing inflammatory burden.

# Weaknesses:

The manuscript presents valuable contributions to tubulitis scoring; however, several areas could benefit from clarification and improvement:
1. The introduction defines tubulitis grading on a 4-point scale (T0: none, T1: 1–4, T2: 5–10, T3: >10), while the results section presents different criteria (ŷ = |Tinflamed| < 2, 1 < maxt ct < 5, 5 ≤ maxt ct ≤ 10, maxt ct > 10). It would be helpful to clarify this discrepancy to ensure consistent understanding throughout the manuscript.
2.	The comparison in Table 2 with existing methods (Cooper et al.) should be done by retraining their method on in-house dataset to ensure fairness as results were obtained on different datasets with varying complexity levels.
3.	For the modular baseline comparisons, please clarify if both models use a same or different segmentation network for tubulitis segmentation, particularly since both methods appear to have different numbers of classes.
4.	The methodology section would benefit from additional technical details, particularly clarification of the phrase "compute forward and backward attention independently."
5.	Key parameters such as magnification level and the top-attended parameter value could be specified to enhance reproducibility.
6.	The rationale for selecting TransMIL could be strengthened by explaining why relationships between different tubulitis regions are expected, given that TransMIL assumes relationships between WSI regions.
7.	Additional details on ground truth annotation generation would be valuable, especially considering the noted inter-reader variability among pathologists (κ = 0.17).
8.	The description of CLAM as a method that "treats patches independently and limits its bag-level representation to top-k selected patches, potentially missing spatially distributed inflammatory patterns" only reviews on top-k selected patches might need correction as it uses entire bag for classification.